# A Validation and Cost-Analysis Study of a Targeted School-Based Dental Check-Up Intervention: Children’s Dental Program

**DOI:** 10.3390/children7120257

**Published:** 2020-11-26

**Authors:** Tan Minh Nguyen, Bradley Christian, Sajeev Koshy, Michael Vivian Morgan

**Affiliations:** 1Deakin Health Economics, Institute of Health Transformation, Faculty of Health, Deakin University, Waurn Ponds, VIC 3216, Australia; 2Community Dental Program, Peninsula Health, Frankston, VIC 3199, Australia; 3Coburg Hill Oral Care, Coburg North, VIC 3058, Australia; 4Dentistry and Oral Health, La Trobe Rural Health School, La Trobe University, Bundoora, VIC 3552, Australia; b.christian@latrobe.edu.au; 5Centre for Oral Health Outcomes & Research Translation (COHORT), School of Nursing & Midwifery, Western Sydney University, Sydney, NSW 2751, Australia; 6Dental Health Services Victoria, Carlton, VIC 3053, Australia; sajeev.koshy@dhsv.org.au; 7Faculty of Dentistry, University of Otago, Dunedin 9016, New Zealand; mike.morgan@otago.ac.nz

**Keywords:** dental care for children, school health services, oral health, preventive health services, costs and cost analysis

## Abstract

Background: Limited evidence exists to inform best practice approaches to implement school-based dental screening to address child retention via referral for dental services. This research tested the null hypothesis that a targeted school-based dental check-up program (intervention) has a 75% child retention rate for public dental care (H_0_ = 0.75). Methods: A prospective non-randomised controlled trial was conducted with a convenience sampling approach in metropolitan Melbourne, Australia. Children in the intervention group were recruited from two preschools and two primary schools from a low socioeconomic area. Children in the standard care group were recruited from the local public dental service. Statistical analysis was performed using Stata IC Version 12. Results: Children in the intervention (45%) were significantly less likely to have never had a dental check-up compared to standard care (20%) (*p* < 0.001). There was no significant difference for the child retention rate for the intervention group when compared against the null hypothesis (*p* = 0.954). The total society costs were AU$754.7 and AU$612.2 for the intervention and standard care groups, respectively (*p* = 0.049). Conclusions: This validation study provides evidence that a targeted school-based dental check-up program can achieve a 75% child retention rate and should be considered for program expansion.

## 1. Introduction

Untreated childhood dental caries in the deciduous dentition is the 10th most prevalent health condition affecting 9.0% of the global child population in 2010 and its prevalence has remained relatively unchanged [1]. In Australia, 27.1% of children aged 5–10 have untreated dental caries in the deciduous dentition and 10.9% of children aged 6–14 years have untreated dental caries in the permanent dentition [2]. Children less than ten years of age experience a high burden of potentially preventable hospitalisations due to poor oral health in comparison to all other age groups [3].

Despite most Australian children being eligible to receive free or low-cost dental care within the public sector, the dental care utilisation rate among children has remained proportionately low. Population oral health surveillance data shows that 57.4% of children aged 13–14 years have never visited a dental practitioner before age 5 [2]. Among children aged 2–17 years from low-income families, only 37.9% of children eligible under the Child Dental Benefits Scheme (CDBS) have claimed benefits under the program [4]. The CDBS is a means-tested Australian federal government program, which provides up to AU$1000 value of dental services over a two-year period.

Traditionally, school dental screening programs have been implemented to address oral health inequities, particularly for children from low-income families who are at increased risk for dental caries. There is limited high-quality evidence to support this type of public health intervention in increasing dental care utilisation, and for its effectiveness to improve childhood oral health [5,6]. Several studies have reported that the proportion of children returning for dental treatment via referral from school dental screening programs can be up to 50% [7].

Different approaches to school dental screening have been trialled. It can yield high levels of child participation at the start, but the child retention rate for dental treatment via referral dropped significantly from 81% to 40% [8]. From a health economics perspective, the opportunity costs lost where there is limited benefit to society when compared to an alternative public health intervention is an important consideration for resource allocation and policy decision-makers.

However, school settings remain an important environment to support childhood health and wellbeing. They include the utilisation of school nurses for physical and mental health support [9,10] and public health policies for school-based vaccinations [11]. Given that family-level characteristics have an influential role in children’s oral health [12], interventions that include the primary carer in the oral health management of the child within school settings should be further investigated.

Previous work has demonstrated that a pilot school-based dental check-up program could achieve a child retention rate of 74% [7], and that the program was less costly and more effective than standard care [13]. In contrast to school dental screening, where the primary carer is often absent, a targeted school-based dental check-up program is a point of difference since it is mandatory for the primary carer to accompany their child at the school premise.

There are other important psychosocial considerations that support why maintaining optimal oral health to children is essential. A previous history of utilising school-based dental services has been associated with a reduction of dental fear and anxiety in adulthood [14]. Improved children’s oral health may also have a role in addressing known associations between untreated dental caries, poor school performance and poor school attendance [15,16].

Healthy weight according to the Body Mass Index (BMI) among children may also be protective against dental caries [17] and impact on a child’s ability to perform everyday activities. However, there is a limited understanding whether there are differences in healthy weight and dental pain experienced by children who are more frequent users of Australian public dental services than those who are not.

The primary aim of this research is to validate the null hypothesis, based on previous published work [13], that a targeted school-based dental check-up program (intervention) can achieve a child retention rate of 75% for public dental care (H_0_ = 0.75) at a two-year follow-up. A secondary objective is to perform a cost-analysis of the intervention from a societal perspective in comparison to standard care at an Australian public dental service. These outcomes are important for health economic modelling and resource allocation decision-making.

## 2. Materials and Methods

### 2.1. Study Design and Target Population

A prospective non-randomised controlled trial was conducted with convenience sampling between 2016 and 2019. This research received Human Research Ethics Approval from the University of Melbourne (HREC 1441662.2).

For the intervention group, children aged 3–12 years who were enrolled in two preschools and two primary schools were invited to participate. The schools were located in a suburb with the lowest 10th percentile for Socio-Economic Index For Areas (SEIFA) index [18] in the geographical region of the City of Whittlesea, 20 km north of Melbourne, Victoria, Australia.

For the standard care group, children from the same age range were recruited by the local public dental service, which provides community healthcare services, including dental services, to residents living in the City of Whittlesea. Children in both groups who have received a dental check-up within the last three months were excluded in this study.

For economic evaluation studies using cost-effectiveness or cost-utility analysis, the focus is to identify whether an alternative intervention should be adopted or not from the cost perspective, rather than to replace an existing intervention. However, other implementation considerations such as health equity may be an important contextual concern for policy decision-makers. Therefore, the sociodemographic profile for both the intervention and standard care groups was explored but not used as a selection criterion.

### 2.2. Sample Size

A minimum of 288 children for each group was required to test the null hypothesis (H_0_ = 0.75; 95% confidence; ±5% proportion range) [19]. Child retention was defined as whether a child in the intervention group who required referral had attended at least an additional dental appointment at the local public dental service by the end of the two-year follow-up.

As a comparator, child retention in the standard care group was determined if the child attended at least an additional dental appointment for dental treatment. In some circumstances, dental treatment for children was provided at the first dental check-up visit due to the convenience of being present at the dental clinic.

### 2.3. Training and Calibration

To minimise the inconvenience to potential participants, calibration for the dental examinations were performed ex vivo rather than in vivo. A combination of 16 tooth surfaces from 16 extracted deciduous and permanent teeth, sterilised and placed on an individual cast, were scored by a gold standard examiner (H.C.).

Three dental examiners (T.M.N., M.N. and C.E.) were trained and calibrated using the modified International Caries Detection and Assessment System II (ICDAS) tool. Code 1 was excluded since the availability of triplex air to dry and assess the teeth for the intervention group was not available. Untreated dental caries at baseline were diagnosed at d/D_3_-d/D_6_ level, i.e., being diagnosed at least to the severity of cavitated enamel caries for the deciduous and permanent teeth. Dental examiners scored a random combination of the tooth surfaces with the teeth wet on three separate days using a portable headlight, intra-oral mirror, periodontal probe and gauze.

The two-digit recording system was used to establish intra- and inter-rater reliability according to the ICDAS Committee recommendations [20]. The intra-examiner Kappa score ranged between 0.64 and 1.00 and the inter-examiner Kappa score ranged between 0.62 and 0.80 when compared against the gold standard examiner.

T.M.N. and L.N. performed dental check-ups for the intervention group. L.N. and C.E. conducted the dental check-ups for the standard care group. At the two-year follow-up data collection, untreated cavitated dental caries were recorded by uncalibrated dental practitioners within the public dental service for both the intervention and standard care group, assumed to be within dentine. This intentional pragmatic approach minimised the administration burden to the local public dental service and acknowledges the value of a ‘natural’ experiment.

### 2.4. The Intervention

For the intervention group, children were recruited up to four weeks in advance to a scheduled study visit. Primary carers could contact the local public dental service or were contacted upon a signed consent form returned from school. Interpreters were arranged where appropriate using the school’s existing interpreter services.

The dental check-up involved children seated in an age-appropriate chair. Calibrated dental examiners performed the dental check-up using portable dental equipment following relevant infection control procedures.

It was mandatory that the primary carer attended the child’s school-based dental check-up appointment. Children recommended for further dental treatment, a ‘positive’ referral, had follow-up calls made by the local public dental service to arrange an additional dental appointment for dental treatment. Alternatively, primary carers could actively call the local public dental service. Children having a ‘positive’ referral received dental treatment following standard care. Children with a ‘negative’ referral were placed on six-month recall for a routine dental check-up.

### 2.5. Standard Care

Children recruited in the standard care group occurred when they had a routine dental check-up scheduled at the local public dental service. Children who did not require dental treatment were placed on either a 12-month, 18-month or 24-month recall for a routine dental check-up according to the local public dental service clinical guidelines.

### 2.6. Data Collection

A semi-structured questionnaire assisted by dental support staff was administered to participants and recorded the social and dental history (Appendix A
Figure A1). They included:Previous history of the last dental visit,Household income,Card-holder status (eligibility for subsidised publicly funded healthcare),Eligibility for the CDBS, andCultural background classified using the Australian Standard Classification of Cultural and Ethnic Groups (ASCCEG) [21].

Quality of life data were collected for 7–12-year-olds from both groups using the EQ-5D-Y self-rated questionnaire [22] (Appendix A
Table A1 and Figure A2). The participants’ self-rated general health was recorded using the EQ visual analogue scale (EQ VAS). The health profiles and EQ VAS data were reported using the recommended guidelines [22].

The EQ-5D-Y was planned to be used for economic evaluation. However, this is not possible at the time of writing [23]. The results of the EQ-5D-Y are presented to provide a comparison of the perceived general health quality of life between the intervention and standard care groups.

Untreated dental caries was recorded as being at least cavitated at baseline (ICDAS ≥ 3) (Appendix A
Figure A3). In-depth analysis of dental services received by children and the teeth health transition states will be reported in another paper focusing on the health economics perspective.

The child’s height and weight were measured using a measure tape and household weight scale. Data input for the child’s height and weight were classified using an Australian online healthy weight calculator using the US Centres for Disease Control and Prevention’s BMI-for-age charts [24] (Appendix A
Table A2). A category for ‘Unhealthy Weight’ was developed as the combined weight status of ‘Below Healthy Weight’, ‘Above Healthy Weight’ and ‘Well Above Healthy Weight’.

For the two-year follow-up evaluation, a record audit was performed by TMN regarding:Date of birth,Sex,Principle place of residence classified using the SEIFA index [18],Dental referral/treatment required after a dental check-up,Whether the child received any restorations or extractions for dental caries,The status if the child’s dental treatment plan was completed,Dental treatment time,Number of dental visits,The number and type of dental treatment services provided according to the Australian Schedule of Dental Services and Glossary [25], andThe prevalence of untreated and treated dental caries (in units of teeth).

### 2.7. Cost-Analysis

The method to calculate the total society costs, which includes the cost components for travel, dental treatment, and opportunity costs lost, was based on previously published work [13].

The travel cost was estimated as the sum of the travel time incurred to attend and return home from dental appointments and the associated travel distance using recommended rates at AU$0.66 per kilometre [26], assuming travel by car. The travel costs were assumed to be zero where the child’s principle of residence is in the same suburb as the local public dental service, or if the first visit was at a school-based dental check-up for children in the intervention group.

Dental treatment costs were derived from the dental record audit and its costs estimated using the CDBS fee schedule [27]. Any dental treatment services not chargeable under the CDBS, although chargeable via the public dental service, was estimated using the Dental Weighted Activity Unit (DWAU), where 0.09 DWAU is equal to AU$52.65 for a comprehensive oral examination.

The opportunity costs lost, which included travel and dental treatment time, was estimated using the potential mean hourly work rate of productivity loss of the primary carer. For an hourly work rate not stated or work status of being unemployed, the national minimum wage for adults at AU$17.7 per hour was applied [28].

For the cost-analysis, there is a chargeable service fee for travel to provide dental services. In reality, the travel costs from a healthcare perspective to perform the intervention would be significantly less. Therefore, one-way sensitivity analysis was performed whereby only one service fee for travel to provide services per study visit day was applied, rather than multiple units of the service fee for travel for each child in the intervention group.

### 2.8. Data Analysis

All data was entered and cleaned using Excel 365 (Microsoft Corporation™, Washington, DC, USA). Summary descriptive statistics, two-group mean-comparison tests (*t*-tests) and two-group proportion tests (*z*-tests) were performed using Stata IC Version 12 (Statacorp™, College Station, TX, USA), where statistical significance was determined at *p* < 0.05. A one-sample proportion test (*z*-test) was performed to test the null hypothesis (H_0_ = 0.75) for the child retention rate.

## 3. Results

A total of 331 children participated: 168 children in the intervention and 163 children in standard care. Table 1 and Table 2 describes the summary statistics for the social circumstances and dental history, with a mean follow-up period of 2.4 years (±0.3 standard deviation (SD)) and non-participation rate of 51% for the intervention group. The mean age for the intervention and standard care groups was 6.2 years (±2.8 SD) and 7.3 years (±2.8 SD), respectively (*p* < 0.001). The participation recruitment process fell short of the target sample size of 288 per group (<58.3%), and thus affected the study’s statistical power.

There were several characteristic dental and social history variables that had no significant difference: sex (*p* = 0.068), household income (*p* = 0.231), card-holder status (*p* = 0.138), CDBS eligibility (*p* = 0.962), missed school due to dental pain (*p* = 0.933) and frequency of missing school due to pain (*p* = 0.067).

There were significant differences for the child’s principle place of residence according to the SEIFA index (*p* < 0.001), if children never had a dental check-up (*p* < 0.001) and whether the child has previously used the local public dental service (*p* < 0.001). For the intervention group, there was no statistically significant difference for the child retention rate (H_0_ = 0.75; *p* = 0.954).

The EQ-5D-Y health profiles and EQ VAS data are reported in Table 3. The number of children who met the inclusion criteria was 40 for the intervention and 60 for standard care. There were no significant differences for the health dimensions of ‘Mobility’ (*p* = 0.078), ‘Doing Usual Activities’ (*p* = 0.547), ‘Feeling Worried, Sad or Unhappy’ (*p* = 0.078). There were significant differences for the health dimensions of ‘Looking After Myself’ (*p* = 0.021) and ‘Having Pain or Discomfort’ (*p* = 0.010). There was a higher self-rated EQ VAS score for the standard care group than the intervention group of 90.7 (±13.2 SD) and 81.9 (±18.0 SD), respectively.

All other comparisons are reported in Table 4. In terms of weight status, significant differences between ‘Healthy Weight’ and ‘Unhealthy Weight’ were found (*p* = 0.005). There were no significant differences for untreated dental caries at d/D_3_-d/D_6_ level at baseline or untreated dental caries in dentine at the two-year follow-up. There were significant differences for non-cavitated dental caries for the deciduous (*p* = 0.001) and permanent (*p* = 0.010) teeth at baseline. Significant differences were observed for all cost categories except for dental treatment costs (unadjusted).

## 4. Discussion

This study demonstrated consistent findings from our previous work that the intervention increases utilisation of dental services for children who never had a dental check-up [13] and achieved the target null hypothesis. It appears that a school-based dental check-up intervention may be more effective than traditional school-based dental screening to address child retention rates.

There is some evidence the CDBS Australian federal dental program may not have a major impact on increasing the utilisation of dental services in the target population. Corroborating observations note that many children in the intervention group never had a dental check-up (*p* < 0.05) compared to standard care. Government reviews of the program indicate that there is a steady increasing rates of dental services utilisation, but it remained proportionally low, from 29.5% to 37.9%, between 2014 and 2018 [4].

There were similar self-reported symptoms for missed school attendance due to dental pain and frequency by children in both groups. However, the social impacts on general health were more apparent for older children in the intervention group for the health dimensions of ‘Looking After Myself’ and ‘Having Pain or Discomfort’. These observations were also supported with lower self-rated health profiles from the EQ VAS scale.

Previous research reported that dental caries is associated with both high and low BMI [17]. Similarly, in our study, we found that there was a significant difference for healthy weight and unhealthy weight status. This could be explained by the higher levels of untreated cavitated dental caries in the deciduous teeth for children in the intervention group since an unhealthy diet is a major risk factor for dental caries risk [29].

Our research identified that children in the intervention group were more likely to require dental treatment (*p* < 0.05), which was not observed in our previous study. However, the cost-analysis is consistent with the observed lower total society costs for the intervention group compared to standard care [13]. The findings have important implications from a health economics and public policy perspective. Firstly, the implementation of subsidised publicly funded dental care, such as the CDBS, may not address dental care utilisation issues, particularly, for children from low-income families.

School-based dental programs, such as those observed in our study, have several advantages. It can address inequities associated with the utilisation of dental services for children. This may lead to a reduction of school absenteeism due to dental pain, improve quality of life, and reduce the prevalence of unhealthy weight if children become regular users of dental services. However, the components of the program would need to be carefully considered.

We know there is enough evidence that school dental screening may have limited benefits to society. Should school-based dental programs focus on preventive interventions or should it include a dental treatment component as well? Our work presented here is a replication study, which demonstrated that the intervention could be integrated well using the existing local public dental service.

Other school-based preventive dental programs such as the application of fluoride varnish and fissure sealants have demonstrated clinical and economic benefits [30,31,32,33,34], with some evidence that fluoride varnish programs may be more cost-effective than fissure sealants [34]. Our study can be enhanced feasibly by incorporating fluoride varnish applications, which is efficacious to prevent dental caries [35] and is cost-effective from an Australian healthcare perspective [36].

### Limitations

In resource-constrained Australian public dental services, school-based preventive dental programs are likely to yield larger health and economic benefits at the population-level than school-based dental programs that have a dental treatment component, relative to the costs incurred. It should be noted that our intervention may not be translatable to other countries due to differences in healthcare systems.

The findings of this study should be interpreted with caution, with regards to the study’s suboptimal sample size and its effect on statistical power. The intervention described used a passive recruitment approach, which has resulted in lower participation rates than estimated. This observation was also found in our previous work [13]. However, the consistent findings of the primary and secondary aim of this replication study are indicative of relative validity.

One of the robust advantages of this study was the ability to observe changes to untreated cavitated dental caries and the associated total society costs associated with local public dental services provided to children mimicked a ‘natural’ experiment, i.e., there were not strict guidelines required by dental practitioners to manage the child’s oral health.

Therefore, any observed effects on dental caries outcomes can be generalisable to real-world scenarios. It is plausible that having mandatory requirements for the primary carers to attend their child’s school-based dental check-up appointment influenced high child retention rates. However, there are several important study weaknesses.

Firstly, untreated cavitated dental caries outcomes were censored and assumed to be diagnosed within dentine. Thus, the total observed outcomes of dental caries at the two-year follow-up are likely to be underestimated since the diagnoses were not performed by calibrated dental practitioners. Our study was not focused on dental caries outcomes, but we recognise that a future study should address this methodological flaw.

Secondly, we do not know if children who participated in the study may have sought dental treatment externally from the local public dental service being referred to. Consequently, the total society costs incurred could be underestimated, especially if a child required dental treatment under general anaesthetic, where the costs for a hospital admission is estimated to be between AU$2500 to AU$8000 in the private sector, or the mean cost of AU$3029 in the public sector [37].

Thirdly, when healthcare services are free or low cost, which was the case for children in this study, there is a concern of increased consumption of healthcare compared to when healthcare services have high out-of-pocket expenses, commonly referred in health economics as a ‘moral hazard’ [38]. But public dental services generally have societal pressure to provide dental care to more people in the eligible population to reduce waiting lists, thereby mitigating this risk.

A targeted school-based dental check-up program may further reduce costs and improve efficiencies by minimising low-value care that is often provided in standard care. For example, preventive services that are commonly provided at a routine dental check-up at a dental clinic includes oral prophylaxis and scaling, which has been shown to have limited health benefits [39]. Families would also need to make a trade-off between attending additional dental appointments for their child to receive dental treatment or going to work.

## 5. Conclusions

This validation study demonstrated that a targeted school-based dental check-up program has a child retention rate of 75% and increases the utilisation of dental services for children from low-income families. Children receiving dental treatment services from the intervention or standard care group had similar untreated cavitated dental caries outcomes at baseline and at the two-year follow-up. Additional investigations are required to evaluate the health economic impact of a school-based dental check-up program compared to standard care to inform whether the intervention should be implemented more extensively.

## Figures and Tables

**Table 1 children-07-00257-t001:** Summary statistics, cultural background, demographic and social profile of children.

Demographics and Social History	Standard Care (*n* = 163) *n* (%)	Intervention (*n* = 168) *n* (%)	*p*-Value
Mean Age (SD) (y)	7.3 (2.8)	6.2 (2.8)	0.001 *
Sex			
Female	92 (56%)	78 (46%)	Reference
Male	71 (44%)	90 (54%)	0.068
Cultural Background (ASCCEG Classification)			
Not Stated	5 (3%)	9 (5%)	-
Multiracial	22 (13%)	15 (9%)	-
Oceanian	60 (37%)	20 (12%)	-
Southern and Eastern European	14 (9%)	9 (5%)	-
North-West European	12 (7%)	3 (2%)	-
North-East Asian	2 (1%)	8 (4%)	-
South-East Asian	5 (3%)	9 (5%)	-
Southern and Central Asian	27 (17%)	46 (27%)	-
North African and Middle Eastern	16 (10%)	44 (26%)	-
Sub-Saharan African	0 (0%)	5 (3%)	-
Principal Place of Residence (SEIFA Classification)			
1–5	47 (29%)	153 (91%)	Reference
6–10	116 (71%)	15 (9%)	0.001 *
Household Income			
Not Stated	24 (14%)	36 (21%)	-
<$50,000	71 (44%)	77 (46%)	Reference
≥$50,000	68 (42%)	55 (33%)	0.231
Card-holder Status			
Yes	88 (54%)	77 (46%)	Reference
No	75 (46%)	91 (54%)	0.138
CDBS Eligibility			
Yes	84 (52%)	87 (52%)	Reference
No	79 (48%)	81 (48%)	0.962
Has Private Health Insurance			
Yes	34 (21%)	27 (16%)	Reference
No	129 (79%)	141 (84%)	0.261

SD = standard deviation; ASCCEG = Australian Standard Classification of Cultural and Ethnic Groups; SEIFA = Socio-Economic Index For Areas; CDBS = Child Dental Benefits Schedule; * statistically significant *p* < 0.05.

**Table 2 children-07-00257-t002:** Summary statistics of the children’s dental history.

Dental History	Standard Care (*n* = 163) *n* (%)	Intervention (*n* = 168) *n* (%)	*p*-Value
Duration Since Last Dental Check-up			
Never	33 (20%)	76 (45%)	Reference
Had previously a dental check-up	130 (80%)	92 (55%)	<0.001 *
<6 Months	26 (16%)	36 (21%)	-
7–12 Months	80 (49%)	36 (21%)	-
13–24 Months	16 (10%)	11 (6%)	-
>24 Months	8 (5%)	9 (5%)	-
Is a Previous Patient with Dianella Plenty Valley Health			
Yes	97 (60%)	49 (29%)	Reference
No	66 (40%)	119 (71%)	<0.001 *
Previous Dental Service Type			
Not Applicable	29 (18%)	77 (46%)	-
Public	100 (61%)	54 (32%)	-
Private	29 (18%)	31 (18%)	-
Overseas	5 (3%)	6 (4%)	-
Missed School Due to Dental Pain			
Yes	14 (9%)	14 (8%)	Reference
No	149 (91%)	154 (92%)	0.933
Frequency of Missing School Due to Dental Pain			
Never	149 (91%)	154 (92%)	-
Sometimes	3 (2%)	0 (0%)	Reference
Frequently	11 (7%)	14 (8%)	0.067

* statistically significant *p* < 0.05.

**Table 3 children-07-00257-t003:** The EQ-5D-Y health profiles and the EQ Visual Analogue Scale (EQ VAS) data of children in the study.

EQ-5D-Y Dimension		Standard Care (*n* = 60) *n* (%)	Intervention (*n* = 40) *n* (%)	*p*-Value
Mobility	No Problems	58 (97%)	35 (87%)	Reference
	Problems	2 (3%)	5 (13%)	0.078
Looking After Myself	No Problems	56 (93%)	31 (77%)	Reference
	Problems	4 (7%)	9 (23%)	0.021 *
Doing Usual Activities	No Problems	56 (93%)	36 (90%)	Reference
	Problems	4 (7%)	5 (10%)	0.547
Having Pain or Discomfort	No Problems	52 (87%)	26 (65%)	Reference
	Problems	8 (13%)	14 (35%)	0.010 *
Feeling Worried, Sad or Unhappy	No Problems	52(87%)	29 (72%)	Reference
	Problems	8 (13%)	11 (28%)	0.078
EQ VAS Data				
Mean (SD)		90.7 (13.2)	81.9 (18.0)	-
25th Percentile		85	70	-
50th Percentile		97	82.5	-
75th Percentile		100	100	-

* statistically significant *p* < 0.05.

**Table 4 children-07-00257-t004:** The children’s weight and untreated dental caries status and dental care characteristics of children in the study.

Health Status and Dental Care Characteristics	Standard Care (*n* = 163) *n* (%)	Intervention (*n* = 168) *n* (%)	*p*-Value
Weight Status			
Unknown	3 (2%)	8 (5%)	-
Below Healthy Weight	33 (20%)	22 (13%)	-
Above Healthy Weight	13 (8%)	26 (15%)	-
Well Above Healthy Weight	17 (10%)	40 (24%)	-
Healthy Weight	97 (60%)	72 (43%)	Reference
Unhealthy Weight	63 (39%)	88 (55%)	0.005 *
Untreated Dental Caries Baseline (Modified ICDAS II [20]			
d2	2.40 (3.1)	1.40 (2.2)	0.001 *
d3–6	0.93 (1.8)	1.39 (2.5)	0.056
D2	1.07 (2.1)	0.57 (1.3)	0.010 *
D3–6	0.19 (0.6)	0.18 (0.7)	0.873
Untreated Cavitated Dental Caries at Follow-Up			
d	0.38 (1.1)	0.64 (1.7)	0.094
D	0.04 (0.2)	0.05 (0.3)	0.590
Referred for Dental Treatment			
Yes	75 (46%)	104 (62%)	Reference
No	88 (54%)	64 (38%)	0.004 *
Received Surgical Dental Treatment			
Yes	46 (28%)	41 (24%)	Reference
No	117 (72%)	127 (76%)	0.430
Child Retention Rates at Follow-Up			
Yes	71 (95%)	76 (75%)	Reference
No	4 (5%)	25 (25%)	0.001 *
Null Hypothesis Test (Intervention Only)	-	75%	0.954
Dental Treatment Time (SD) (min)	115.1 (120.6)	79.6 (79.1)	0.002 *
Number of Dental Visits (SD)	3.2 (2.5)	2.6 (1.7)	0.011 *
Travel Costs (SD) (AU$)	19.0 (27.3)	8.0 (10.9)	0.001 *
Dental Treatment Time Cost (SD) (AU$)	40.4 (41.8)	28.0 (28.4)	0.002 *
Dental Treatment Cost (SD) (AU$)			
Unadjusted	695.4 (627.8)	585.1 (516.5)	0.081
Adjusted	-	517.6 (517.4)	0.005 *
Total Society Cost (SD) (AU$)			
Unadjusted	754.7 (674.9)	621.2 (547.8)	0.049 *
Adjusted	-	533.8 (548.8)	0.003 *

ICDAS II = International Caries Detection and Assessment System (ICDAS II); d = decayed deciduous tooth; D = decayed permanent tooth; AU$ = Australian dollars; * statistically significant *p* < 0.05.

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
