# Peer review of "A Validation and Cost-Analysis Study of a Targeted School-Based Dental Check-Up Intervention: Children’s Dental Program"

_children, 2020, doi:10.3390/children7120257_

Round 1

Reviewer 1 Report

Thank you for asking me to review this paper. The paper is well wriiten and interesting however I have some more major concerns with the conclusions drawn given the study design which I will go over. I have also noted some minor amendments below.

The discussion / conclusions state that a school based dental check-up program increases the utilisation of dental services but you were comparing this against those already going to the dentist and using a service. Why not recruit standard groups from schools who are not referred and compare this – (again a randomised trial would give more confidence in the results). I’m not certain from this study design you can say that the dental check-up increased the use of dental services as many may have attended the dentist anyway. Are you basing this solely on those who had never attended and then went to the dentist, after stating they should attend (if so this isn’t clear in your results and tables)? Again why was this not compared against a standard control, who could have been given a leaflet for example about oral health. If you then noted a difference in attendance this could be attributed to the dental screening. As you have recruited from two very different settings I’m not too sure what the standard care is showing in this study as a comparison for increase in dental attendance. As discussed in  my comments below – for table 3 you show  75% of those in the intervention group had treatment completed but 62% had treated required. So this doesn’t tell us how many of those actually referred attended for treatment (as from this I assume a proportion who weren’t told to attend went anyway). This is my overall concern with the paper and conclusions drawn as it stands.

Below are some minor comments.

Abstract

Page 1

Line 16 ‘complete’ unclear what you mean here please rephrase

Line 19 change ‘was’ to ‘were’

Introduction

Page 1

Line 32 – the figure you mention is untreated dental caries in deciduous teeth – the % for childhood dental caries overall is much higher – you need to amend to make clear this is untreated dental caries

Page 2

Line 59 – I can’t see in this paper cited that it states 50% attend dental treatment after a referral. I can see this is what the sample size  was based on but I can’t see data to say this was obtained. (reference 12). Unless I have missed this in the results section of this paper. It might be worth to discuss larger studies such as Milsom, Tickle et al who studied 17,500 children for a national dental screening program – the results from this shows a lower number and also the differences according to socio economic groups

Milsom K, Blinkhorn A, Worthington H, Threlfall A, Buchanan K, Kearney-Mitchell P, Tickle M. The effectiveness of school dental screening: a cluster-randomized control trial. J Dent Res. 2006 Oct;85(10):924-8. doi: 10.1177/154405910608501010. PMID: 16998133.

Milsom, K., Threlfall, A., Blinkhorn, A. et al. The effectiveness of school dental screening: dental attendance and treatment of those screened positive. Br Dent J 200, 687–690 (2006). https://doi.org/10.1038/sj.bdj.4813724

Line 70 – ‘frequents’ to ‘frequent’

Page 7 table 1 – why are some of the p values blank – have you not compared previous dental service type for example. Also for frequency of missing school due to pain no p value for never – you are using sometimes as the reference but the intervention has 0 – is there an issue of using this as a reference with 0 responses?

Page 8 table 2 – the column headings don’t fit against the columns *your p value is above what should be the intervention N and %

Table 3 – why are there no p values for the other weight groups? – it looks as though you have just looked at healthy weight vs unhealthy weight but then given % of the breakdown of unhealthy weight? But this is not entirely clear make move the other 3 groups below unhealthy weight and put in a different format to distinguish? You state 75% of those in the intervention group had treatment completed but 62% had treated required – does that mean that you sent a message they should have dental treatment but those that did not need dental treatment wen to the dentist. It would be important to know how many of those referred actually attended the dentists as the others would be on a six month recall and would not have needed to go?

Untreated decay at follow up -  are you assuming this is recorded by dentists into dentine ?- what would be comparable to the baseline

You state in description that untreated dental caries will eb a least cavitated (d3) but show d2 in the table although this is where your standard of care groups has a higher untreated dental caries than your intervention at baseline

What was the cut-off point (how long after referral) was used for if they attended for dental treatment?

For those in the standard care group would they have had their dental treatment completed at the same time the attend for  check up anyway? So you would expect this number to be higher as in they may have received a restoration or other treatment at the time?

Page 10 – Discussion

Line 248 ‘has’ to ‘have’

Page 11

Line 207 – 308 – is there any evidence that you obtain a reduction in unhealthy weight in children who utilise dental services more – there is nothing to back up this sentence.  

Issues of non-participation and failure to meet sample size not fully explored.

Author Response

Review 1

Thank you for asking me to review this paper. The paper is well written and interesting however I have some more major concerns with the conclusions drawn given the study design which I will go over. I have also noted some minor amendments below.

The discussion / conclusions state that a school based dental check-up program increases the utilisation of dental services but you were comparing this against those already going to the dentist and using a service. Why not recruit standard groups from schools who are not referred and compare this – (again a randomised trial would give more confidence in the results). I’m not certain from this study design you can say that the dental check-up increased the use of dental services as many may have attended the dentist anyway. Are you basing this solely on those who had never attended and then went to the dentist, after stating they should attend (if so this isn’t clear in your results and tables)? Again why was this not compared against a standard control, who could have been given a leaflet for example about oral health. If you then noted a difference in attendance this could be attributed to the dental screening. As you have recruited from two very different settings I’m not too sure what the standard care is showing in this study as a comparison for increase in dental attendance.

Response. This study had two distinct groups, the intervention, which is drawn from sample of the target geographical area, and the standard care group, again, recruited from the target geographical area (City of Whittlesea). For clarity, we added at Line 93-96: ‘For the standard care group, children from the same age range were recruited by the local public dental service, which provides a community health services, including dental services, to residents living in the City of Whittlesea. Children in both groups who received a dental check-up within the three months prior to their dental check-up in this study were excluded.’

The evidence tells us that at present only about 38% of the eligible child population utilises dental services. The overarching aim of this program of research is to increase the proportion of the eligible child population that utilises dental services achieve a child retention rate of 75%.  We defined child retention at Line 102:

‘Child retention was defined as whether a child in intervention group who required referral had attended at least an additional dental appointment at the local public dental service by the end of the two-year follow-up.’

As discussed in my comments below – for table 3 you show 75% of those in the intervention group had treatment completed but 62% had treated required. So this doesn’t tell us how many of those actually referred attended for treatment (as from this I assume a proportion who weren’t told to attend went anyway). This is my overall concern with the paper and conclusions drawn as it stands.

Response. We incorrectly described the null hypothesis. The null hypothesis refers to 75% for child retention according to the above definition, noting some children may have not been referred initially, but may develop caries later during the study period. We have revised the null hypothesis at Line 98-101:

‘A minimum of 288 children for each group was required to test the null hypothesis (H0=0.75; 95% confidence; ±5% proportion range) [19]. Child retention was defined as whether a child in intervention group who required referral had attended at least an additional dental appointment at the local public dental service by the end of the two-year follow-up.’

Below are some minor comments.

Abstract

Page 1

Line 16 ‘complete’ unclear what you mean here please rephrase

We have revised this sentence so it now reads:

‘This research tested the null hypothesis that a targeted school-based dental check-up program (intervention) has a 75% child retention rate for public dental care (H0=0.75).’

Line 19 change ‘was’ to ‘were’

 Response. Corrected as advised.

Introduction

Page 1

Line 32 – the figure you mention is untreated dental caries in deciduous teeth – the % for childhood dental caries overall is much higher – you need to amend to make clear this is untreated dental caries

 Response. We have added ‘Untreated’ so it now reads:

‘Untreated childhood dental caries in the deciduous dentition is the 10th most prevalent health condition affecting 9.0% of the global child population in 2010 and its prevalence and remained relatively unchanged [1].’

Page 2

Line 59 – I can’t see in this paper cited that it states 50% attend dental treatment after a referral. I can see this is what the sample size  was based on but I can’t see data to say this was obtained. (reference 12). Unless I have missed this in the results section of this paper. It might be worth to discuss larger studies such as Milsom, Tickle et al who studied 17,500 children for a national dental screening program – the results from this shows a lower number and also the differences according to socio economic groups

Milsom K, Blinkhorn A, Worthington H, Threlfall A, Buchanan K, Kearney-Mitchell P, Tickle M. The effectiveness of school dental screening: a cluster-randomized control trial. J Dent Res. 2006 Oct;85(10):924-8. doi: 10.1177/154405910608501010. PMID: 16998133.

Milsom, K., Threlfall, A., Blinkhorn, A. et al. The effectiveness of school dental screening: dental attendance and treatment of those screened positive. Br Dent J 200, 687–690 (2006). https://doi.org/10.1038/sj.bdj.4813724

 Response. The reference for 50% is not available electronically, however, we have incorporated some discussion about the paper by Milsom and colleagues with a lower follow-up rate via referral at Line 52-56:

‘Different approaches to school dental screening have been trialed. It can yield high levels of child participation at the start, but the child retention rate for dental treatment via referral dropped significantly from 81% to 40% [8]. From a health economics perspective, the opportunity costs lost where there is limited benefit to society when compared to an alternative public health intervention, is an important consideration for resource allocation and policy decision-makers.’

Line 70 – ‘frequents’ to ‘frequent’

Response. Correction has been made.

Page 7 table 1 – why are some of the p values blank – have you not compared previous dental service type for example.

Response. The aims of this study only focused on 2 objectives, testing the 1) intervention against the null hypothesis and 2) performed a cost-analysis for future economic evaluation. Therefore, several variables were not subjected to statistical analysis because they are reported to provide a profile of the groups only. We were only interested how long ago was the last dental check-up since this more representative of an individual being a regular user of dental service, rather than an individual who only attends for a dental emergency.

Also for frequency of missing school due to pain no p value for never – you are using sometimes as the reference but the intervention has 0 – is there an issue of using this as a reference with 0 responses?

Response. For the ‘Frequency of Missing School Due to Dental Pain’ variable, there did not appear to be an error for statistical analysis if the Intervention had 0.

Page 8 table 2 – the column headings don’t fit against the columns *your p value is above what should be the intervention N and %

 Response. We have formatted all table headings for ease of read as recommended.

Table 3 – why are there no p values for the other weight groups? – it looks as though you have just looked at healthy weight vs unhealthy weight but then given % of the breakdown of unhealthy weight? But this is not entirely clear make move the other 3 groups below unhealthy weight and put in a different format to distinguish?

Response. A previous review showed an associated between dental caries and high BMI and low BMI. Therefore, we have only made a comparison between unhealthy weight and healthy weight, and defined the unhealthy weight at Line 167-169:

‘A category for ‘Unhealthy Weight’ was developed as the combined weight status of ‘Below Healthy Weight’, ‘Above Healthy Weight’ and ‘Well Above Healthy Weight’. We have re-ordered the healthy weight categories in Table 3.

You state 75% of those in the intervention group had treatment completed but 62% had treated required – does that mean that you sent a message they should have dental treatment but those that did not need dental treatment wen to the dentist. It would be important to know how many of those referred actually attended the dentists as the others would be on a six month recall and would not have needed to go?

Response. Our study was interested in child retention for the intervention in comparison to the null hypothesis and used this as the primary outcome of the study. We defined child retention as per a previous comment at Line 99-101:

‘Child retention was defined as whether a child in intervention group who required referral had attended at least an additional dental appointment at the local public dental service by the end of the two-year follow-up.’

Untreated decay at follow up -  are you assuming this is recorded by dentists into dentine ?- what would be comparable to the baseline

Response. We reported that dental caries diagnosed by uncalibrated dental practitioners and assumed to be at dentine level. We have clarified this at Line 120-122:

‘At the two-year follow-up data collection, untreated cavitated dental caries was recorded by uncalibrated dental practitioners within the public dental service for both the intervention and standard care group, assumed to be at dentine level.’

You state in description that untreated dental caries will eb a least cavitated (d3) but show d2 in the table although this is where your standard of care groups has a higher untreated dental caries than your intervention at baseline

Response. We have clarified the differences for untreated dental caries at d/D3-d/D6 level at Line 112: ‘Untreated dental caries at baseline were diagnosed at d/D3-d/D6 level.’

What was the cut-off point (how long after referral) was used for if they attended for dental treatment?

Response. As per previous comment, we have now defined child retention rate, and updated relevant statistical analysis according to this definition.

For those in the standard care group would they have had their dental treatment completed at the same time the attend for  check up anyway? So you would expect this number to be higher as in they may have received a restoration or other treatment at the time?

Response. We agree. We reviewed and ran additional data analysis reflecting on the definition of child retention rate. As reported in Table 4, proportionally, children referred in the standard care group was lower at 46% compared to 62%. In addition, the child retention rate was higher at 95% compared to 75%, perhaps because standard care service users may have greater health literacy. However, was have not discussed since this was not our primary or secondary aim.

Page 10 – Discussion

Line 248 ‘has’ to ‘have’

 Response. Corrected as advised.

Page 11

Line 207 – 308 – is there any evidence that you obtain a reduction in unhealthy weight in children who utilise dental services more – there is nothing to back up this sentence.  

 Response. Our intervention did not explore the effects of dental services and healthy weight status. We only observed that there was a higher proportion of unhealthy weight found in the intervention group, which also had higher levels of untreated cavitated dental caries in the deciduous teeth.

Issues of non-participation and failure to meet sample size not fully explored.

Response. We agree and have addressed this at Line 304-308:

‘The findings of this study should be interpreted with caution, with regards to the study’s suboptimal sample size and its effect on statistical power. The intervention described used a passive recruitment approach, which has resulted in lower participation rates than estimated. This observation was also found in our previous work [13]. However, the consistent findings of the primary and secondary aim of this replication study is indicative of relative validity.’

Reviewer 2 Report

The project is described in detail, sometimes hard to read due to the presence of much information. It would be suggested to summarize the passages of material and method to make it clearer and schematic, to help the reader in the comprehension of the design of the study.

In Introduction change the position of the paragraph from 66 line to 73 line to the beginning of the introduction before the “school dental screening” consideration. The aims here reported could be implemented in order to have as a goal the description of the two studied population because are deeply investigated by the authors during all phases of the project.

In material and methods, the questionnaire (appendix B) seems not adequate to study children general health. Please explain in the discussion the motivation of the choice of this questionnaire. In training and calibration, the teeth calibration was performed in an ex vivo and not in vivo condition, where the natural colour and hydration of the teeth result altered.

In the results, I suggest removing p value if not statistically significant to highlight just the significant results. Please, reorganize Table 2, because columns are shifted and removed p value if it is not statistically significant. Then in all table, please divide with lines in different categories.  Furthermore, is reported by the Author that the number of treated subjects is not in correlation with the result of the power analysis. This point might be deeply explained in the discussion.

The discussion must be more focalized on the answer obtained from the demonstrated hypothesis/aims, dividing it in different steps.

A limitation of the study could be that the project results to have a scientific utility particularly for Australian dental system, sometimes it seems difficult to be applied in other countries.

In the References, revise and make uniform the references as reported in the author guidelines. 

Author Response

Reviewer 2

The project is described in detail, sometimes hard to read due to the presence of much information. It would be suggested to summarize the passages of material and method to make it clearer and schematic, to help the reader in the comprehension of the design of the study.

Response. We have re-ordered details of the study methods for better flow and made changes where appropriate for simplicity. We have deleted information that is assumed knowledge research protocols.

In Introduction change the position of the paragraph from 66 line to 73 line to the beginning of the introduction before the “school dental screening” consideration. The aims here reported could be implemented in order to have as a goal the description of the two studied population because are deeply investigated by the authors during all phases of the project.

Response. We have moved this paragraph earlier in the Introduction to improve flow and context. Major changes in the Introduction section has been made. Whilst the primary aim is regarding the intervention, a standard care group was required to achieve the secondary objective to perform a cost-analysis in comparison to standard care. We have added a sentence why these aims were important at Line 78-83:

‘The primary aim of this research is to validate the null hypothesis, based from previous published work [13], that a targeted school-based dental check-up program (intervention), can achieve a child retention rate of 75% for public dental care (H0=0.75) at a two-year follow-up. A secondary objective is to perform a cost-analysis of the intervention from a societal perspective in comparison to standard care at an Australian public dental service. These outcomes are important for health economic modelling and resource allocation decision-making.’

In material and methods, the questionnaire (appendix B) seems not adequate to study children general health. Please explain in the discussion the motivation of the choice of this questionnaire.

Response. The EQ-5D-Y was initially planned to be used a tool in preparation for future economic evaluation. However, the quality of life value sets has not yet been developed at this time of writing. We have expanded on the rationale at Line 156-166:

‘Quality of life data were collected for 7-12-year-olds from both groups using the EQ-5D-Y self-rated questionnaire [22] (Appendix B). The participants’ self-rated general health was recorded using the EQ visual analogue scale (EQ VAS), where the endpoint labels were ‘The best health you can imagine’ and ‘The worst health you can imagine’. The health profiles and EQ VAS data were reported using the recommended guidelines [22].

The EQ-5D-Y was planned to be used for economic evaluation. However, this is not possible at the time of writing [23]. The results of the EQ-5D-Y is presented to provide a comparison of the perceived general health quality of life between the intervention and standard care groups.’

In training and calibration, the teeth calibration was performed in an ex vivo and not in vivo condition, where the natural colour and hydration of the teeth result altered.

Response. We have added the following sentence at Line 106-108 to explain why this method was chosen:

‘To minimise the inconvenience to potential participants, calibration for the dental examinations were performed ex vivo rather than in vivo.’

We have also justified why uncalibrated dental examiners was used at the two-year follow-up at Line 123-124:

‘This intentional pragmatic approach minimised the administration burden to the local public dental service and acknowledges the value of a ‘natural’ experiment.’

In the results, I suggest removing p value if not statistically significant to highlight just the significant results.

Response. We have kept p-values where there is no statistical significance to assist readers that a comparison of the two groups have been made for different variables so they can provide their own critique of the findings.

Please, reorganize Table 2, because columns are shifted and removed p value if it is not statistically significant. Then in all table, please divide with lines in different categories.  

Response. The alignment of Table 2 has been corrected, and division lines for the remaining tables have been added to separate different variables.

Furthermore, is reported by the Author that the number of treated subjects is not in correlation with the result of the power analysis. This point might be deeply explained in the discussion.

Response. This is a point worth noting and discussing. We have added the following content to discuss the concerns of statistical power from a suboptimal sample size at Line 304-308:

‘The findings of this study should be interpreted with caution, with regards to the study’s suboptimal sample size and its effect on statistical power. The intervention described used a passive recruitment approach, which has resulted in lower participation rates than estimated. This observation was also found in our previous work [13]. However, the consistent findings of the primary and secondary aim of this replication study is indicative of relative validity.’

The discussion must be more focalized on the answer obtained from the demonstrated hypothesis/aims, dividing it in different steps.

Response. The general structure of the discussion we have retained. However, we have deleted content where we felt it was irrelevant or repetitive to what was presented in the results. We felt it was important to make a comparison of this study and the previous study we completed prior to identify similarities and differences in the finding as it is a validation study.

A limitation of the study could be that the project results to have a scientific utility particularly for Australian dental system, sometimes it seems difficult to be applied in other countries.

Response. Agreed. We have added a sentence reflecting this at Line 302-303:

‘It should be noted our intervention may not be translatable to other countries due to differences in healthcare systems.’

In the References, revise and make uniform the references as reported in the author guidelines. 

Response. Noted, reviewed and amended according to the journal guidelines.

Reviewer 3 Report

Thank you for the opportunity to review this study. Exploring interventions aimed at reducing oral health disparities in socio-economically disadvantaged populations in important. However, I think there are several areas where this paper can be strengthened. Firstly, the paper would benefit from a close read to detect grammar and syntax issues, especially with (lack of) comma use and awkwardly worded, overly long, sentences. An example is in the background section of the abstract, the sentence beginning "The null hypothesis...", which should be edited for clarity and grammar. Please review throughout.

Introduction: overall, could flow better. The transitions between paragraphs are choppy, and break up the flow of ideas. For example, the last sentence on line 64 is just hanging at the end of the paragraph, and should be better integrated earlier into the paragraph. Paragraph 66-73 seems like an after thought and needs to be edited. Notably, it is the first instance where healthy weight is mentioned, but weight was part of your primary aim, so should therefore play a more prominent part of your introduction.

-Line 50: review for comma use and missing prepositions

Methods: how were the samples matched? Were the dental sites where the standard care population was recruited from also in a low socio-economic area? It's unclear to me whether you are matching the groups, and if not, why not.

Line 106 - some details about sample size calculations would be beneficial, despite them coming from previously published work

Line 147-151 is a run on sentence and is awkwardly structured

Results: did you achieve your sample size to reach power? Earlier you listed needing 288 per group.

Line 233: you have 2 Table 1. Please renumber

Line 236: the formatting is off

Discussion: is well written and clearly presented, with few grammar issues to review.

Author Response

Reviewer 3

Thank you for the opportunity to review this study. Exploring interventions aimed at reducing oral health disparities in socio-economically disadvantaged populations in important. However, I think there are several areas where this paper can be strengthened. Firstly, the paper would benefit from a close read to detect grammar and syntax issues, especially with (lack of) comma use and awkwardly worded, overly long, sentences. An example is in the background section of the abstract, the sentence beginning "The null hypothesis...", which should be edited for clarity and grammar. Please review throughout.

Response: The example provided for grammar corrections has been made:

‘This research tested the null hypothesis that a targeted school-based dental check-up program (intervention) has a 75% child retention rate for public dental care (H0=0.75).’

A review and relevant edits have been made throughout the paper. 

Introduction: overall, could flow better. The transitions between paragraphs are choppy, and break up the flow of ideas. For example, the last sentence on line 64 is just hanging at the end of the paragraph, and should be better integrated earlier into the paragraph. Paragraph 66-73 seems like an after thought and needs to be edited. Notably, it is the first instance where healthy weight is mentioned, but weight was part of your primary aim, so should therefore play a more prominent part of your introduction.

Response. A comparison of healthy weight and unhealthy weight was not our primary aim. We have made consistency reviews for the content to address the primary and secondary aims. We have revised the aims at Line 78-83: ‘The primary aim of this research is to validate the null hypothesis, based from previous published work [13], that a targeted school-based dental check-up program (intervention), can achieve a child retention rate of 75% for public dental care (H0=0.75) at a two-year follow-up. A secondary objective is to perform a cost-analysis of the intervention from a societal perspective in comparison to standard care at an Australian public dental service. These outcomes are important for health economic modelling and resource allocation decision-making.’

-Line 50: review for comma use and missing prepositions

Response: We removed this sentence due to a major rewrite of the paper.

Methods: how were the samples matched? Were the dental sites where the standard care population was recruited from also in a low socio-economic area? It's unclear to me whether you are matching the groups, and if not, why not.

Response: Our research approach adopted a realistic approach in how children may or may not be currently receiving dental care provided by the local public dental service, which provides free or low-cost dental care to all children under 12 years old to residents living in the City of Whittlesea, which is the same geographic area of the intervention group. Clarity is stated at Line 89-96:

‘ For the intervention group, children aged 3-12 years who were enrolled in two preschools and two primary schools were invited to participate. The schools were located in a suburb with the lowest 10th percentile for Socio-Economic Index For Areas (SEIFA) index [18] in the geographical region of the City of Whittlesea, 20 kilometres north of Melbourne, Victoria, Australia.

For the standard care group, children from the same age range were recruited by the local public dental service, which provides a community health services, including dental services, to residents living in the City of Whittlesea. Children in both groups who received a dental check-up within the three months prior to their dental check-up in this study were excluded.’

Line 106 - some details about sample size calculations would be beneficial, despite them coming from previously published work

Response. Details of sample size calculation has been stated but rewritten at Line 98-104:

‘A minimum of 288 children for each group was required to test the null hypothesis (H0=0.75; 95% confidence; ±5% proportion range) [19]. Child retention was defined as whether a child in intervention group who required referral had attended at least an additional dental appointment at the local public dental service by the end of the two-year follow-up.

As a comparator, child retention in the standard care group was determined if the child attended at least an additional dental appointment for dental treatment. In some circumstances, dental treatment for children was provided at the first dental check-up visit.’

Line 147-151 is a run on sentence and is awkwardly structured

Response. This section has been revised as a list:

‘A semi-structured questionnaire assisted by dental support staff was administered to participants and recorded the social and dental history (Appendix A). They included:

  • previous history of the last dental visit
  • household income
  • card-holder status (eligibility for subsidised publicly funded healthcare)
  • eligibility for the CDBS
  • cultural background classified according to the Australian Standard Classification of Cultural and Ethnic Groups (ASCCEG) [21]’

Results: did you achieve your sample size to reach power? Earlier you listed needing 288 per group.

Response: We stated the recruited process fell short of the target sample size and revised the sentence noting issues with statistical power at Line 214-215:

‘The participation recruitment process fell short of the target sample size of 288 per group (<58.3%), thus, and affected the study’s statistical power.’

Line 233: you have 2 Table 1. Please renumber

Response. Errors for all table numbers have been corrected.

Line 236: the formatting is off

Response. Line 236 has been reformatted and corrected.

Discussion: is well written and clearly presented, with few grammar issues to review.

Response. Additional content has been added in the Discussion section reflecting on issues regarding the limitations of the research methods and reviewed for grammar corrections.

Round 2

Reviewer 2 Report

Thank you to have revised the paper following all the suggestions.

The paper results more fluent also if hard follow in all passages because very focused on Australian system. Despite the changes, some more modifications which I will go over, results necessary to improve the article.

Referring the file entitled "children-980830", kindly I suggest to the Authors to correct it as follow:

Line 407 there is a typing error, please change with "healthy"

Table 3 is incorrect in the columns, because under “standard care, intervention care, p value, and EQ-5D-Y Dimension” name of the column, I am not sure that there are the corresponding data. Please check and reformulate the table.

Kindly I suggest to the Authors to explain because in the abstract, they told about “their principle place of residence were in the lowest 50th percentile of socioeconomic disadvantage” concept that in the full-length text seem not be exhaustively reported.

At the line 443 it is not clear what is the meaning of “(code 1 was excluded)”, please define in the text.  Then, the new following phrase "Untreated dental caries at baseline were diagnosed at d/D3-d/D6 level" reports a codification for "d/D3-d/D6" that is not universally used, please insert the citation of classification used in the paper and in the table 4.

Finally in material and methods in particular in the section entitled "Study Design and Target Population" the social and economic criteria that are at the base of the choice of the both groups, are not so immediate for the reader, in particular the criteria for the standard group.

Author Response

Dear Reviewer 2

We appreciate the additional feedback on our work. Please find our responses below. We have also made additional amendments throughout the paper to minimise repetitive content.

Thank you to have revised the paper following all the suggestions.

The paper results more fluent also if hard follow in all passages because very focused on Australian system. Despite the changes, some more modifications which I will go over, results necessary to improve the article.

Referring the file entitled "children-980830", kindly I suggest to the Authors to correct it as follow:

Line 407 there is a typing error, please change with "healthy"

Response. This typo error has been corrected.

Table 3 is incorrect in the columns, because under “standard care, intervention care, p value, and EQ-5D-Y Dimension” name of the column, I am not sure that there are the corresponding data. Please check and reformulate the table.

Response. Thankyou, Table 3 headings has been corrected.

Kindly I suggest to the Authors to explain because in the abstract, they told about “their principle place of residence were in the lowest 50th percentile of socioeconomic disadvantage” concept that in the full-length text seem not be exhaustively reported.

Response. We realised that the concept of addressing socioeconomic disadvantaged was not a primary study aim. Therefore, we have removed the sentence in the Abstract referring to the 50th percentile concept and added the following sentence at Line 30-31, which emphasises our secondary aim:

‘The total society costs were AU$754.7 and AU$612.2 for the intervention and standard care groups, respectively (p=0.049).’

At the line 443 it is not clear what is the meaning of “(code 1 was excluded)”, please define in the text. 

Response. We have removed this text and rephrase this concept using the following sentence at Line 125-126:

‘Code 1 was excluded since the availability of triplex air to dry and assess the teeth for the intervention group was not available.’

Then, the new following phrase "Untreated dental caries at baseline were diagnosed at d/D3-d/D6 level" reports a codification for "d/D3-d/D6" that is not universally used, please insert the citation of classification used in the paper and in the table 4.

Response. We have added the following sentence for clarity at Line 127-128 as well as adding the reference in Table 4:

‘i.e. being diagnosed at least to the severity of cavitated enamel caries for the deciduous and permanent teeth.’

Finally in material and methods in particular in the section entitled "Study Design and Target Population" the social and economic criteria that are at the base of the choice of the both groups, are not so immediate for the reader, in particular the criteria for the standard group.

Response. We have added the following paragraph to support the rationale why the sociodemographic profile was evaluated but not used as a selection criterion at Line 105-110:

‘For economic evaluation studies using cost-effectiveness or cost-utility analysis, the focus is to identify whether an alternative intervention should be adopted or not from cost perspective, rather than to replace an existing intervention. However, other implementation considerations such as health equity may be an important contextual concern for policy decision-makers. Therefore, the sociodemographic profile for both the intervention and standard care groups was explored but not used as a selection criterion.’.